# The Hierarchy of Authorship in the Hermeneutics of the *Daodejing*

## Daniel Sarafinas

Philosophy Department, East China Normal University, Shanghai, 201100, China;
dansarafinas@philo.ecnu.edu.cn

**Abstract:** The question of the authorship of the *Daodejing*, otherwise known as the *Laozi*, is a hotly contested debate, and one's stance on the existence and role of the author can have potential implications for one's interpretation of the text. This paper explores how notions of authorship of a text influence, often unconsciously, a reader's interpretation such that the possible meaning generated within that text becomes limited, reduced, or terminated. Three hermeneutic frameworks, Authorial intentionalism, reader-oriented readings, and intention of the text, are problematized, revealing both how they contribute to the production of meaning, but more importantly how a lack of critical awareness of one's own hermeneutic stance regarding authorship might terminate potential significance. These hermeneutic frameworks are applied to the work of contemporary scholars and translators of the *Laozi* in order to assess how implicit notions of authorship contribute to strengths and weaknesses in interpretations of the *Laozi* as it regards the production of meaning and significance. Being critical in nature, this paper is meant only to reveal how the reader's unreflexive engagement with their attitude toward authorship can lead to problematic results in interpretation and translation of any work in general and the *Laozi* in particular.

**Keywords:** *Laozi*; hermeneutics; authorial intentionalism; reader-oriented text

## 1. Introduction

There are few works of literature whose authorship is more debated than the *Laozi* 老子, particularly within the last fifty years. The traditional view regarding the authorship of the *Laozi* is derived from scattered references in Pre-Qin and Han dynasty texts to a figure referred to as Laozi, Lao Dan 老聃 or 老耽, and Li Er 李耳 often portrayed promoting central themes found in the *Laozi*. The *Lüshi Chunqiu* 呂氏春秋 (Master Lü's Commentary to the Spring and Autumn Annals) for example, describes "Lao Dan esteems *rou* 柔 (suppleness, softness, flexibility)" (Zhang et al. 2011, p. 526), the *Zhuangzi* 莊子 (Book of Master Zhuang) portrays Lao Dan as having recognized the interrelated nature of binaries such that "life and death [is] a single thread," (Ziporyn 2009, p. 35), and in the *Liezi* 列子 (Book of Master Lie) he is depicted directly quoting the *Laozi* "the soft and weak are the disciples of life". (Yang 1979, p. 82). The most commonly cited source attributing the authorship of the *Laozi* to a historical figure is Sima Qian's 司马迁 (145–86 B.C.E.) description of Laozi in his *Records of the Grand Historian* (*Shiji* 史記). Despite this account being written some four hundred years after the historical figure Laozi supposedly existed, this is the widely accepted story of the *Laozi*'s authorship within China and without. For those who do not accept the story of Laozi as author of the titular manuscript, many still maintain that the extant editions of the *Laozi* point to an urtext from which all received editions are ultimately derived. Others are more skeptical of the idea of a single person named Laozi having written the eponymous work, the majority of whom support variations of accretion models regarding the coming-to-be of the received manuscript, or "an edited accumulation of fragments and bits drawn from a wide variety of sources" (Hansen 1992, p. 201). Over the past fifty years, manuscripts related to the *Laozi* have been unearthed, most importantly the Mawangdui

马王堆 silk texts dating to around 168 BCE found in 1973 and the Guodian 郭店 bamboo slips dating to around 300 BCE found in 1993, adding fuel to the debate concerning the historicity of Laozi-the-man and the existence of an ancient urtext.

Disregarding the veracity of claims for Laozi-the-person having written the *Laozi* or the existence of an ancient urtext, this paper will bracket off questions regarding who authored the text or how it was compiled and reflexively direct our awareness towards how one's stance on the existence of Laozi and his "original intention" behind the text affects one's hermeneutic position when interpreting it. With this reflexive attitude, the problem of providing a standard for interpretation as it regards notions of authorship will be explored, utilizing hermeneutic theories as they might be applied to interpretations of the *Laozi*. Examples of interpretations from contemporary scholars in the field will help reveal how the quest to discover the author's original meaning or, the other extreme, the complete rejection of authorial intention and historical context might generate or limit the production of meaning within the *Laozi* and influence the reader's (often unconscious) perception of the boundaries of acceptable interpretations. The first hermeneutic model which directly addresses the problem of authorial intention is that of Roland Barthes.

## 2. Authorial Intentionalism and Its Limits

In his groundbreaking paper *The Death of the Author*, Roland Barthes[1] implores us to reconsider the role of "the Author"[2] in relation to the reader's relationship with "the work". This "Author" is not merely the writer of the work, or the empirical author, but is also what that empirical author represents; the historical time period, political context, and the semiotic relationship between signifier and signified which existed when the empirical author penned the work. According to Barthes, not only is the Author always already removed from the reading of the work because of our lack of access to the empirical author's inner thought, historical context, and the immediacy of the process of semiosis, of but also, as Barthes says:

> Once the Author is removed, the claim to decipher a text becomes quite futile. To give a text an Author is to impose a limit on that text, to furnish it with a final signified, to close the writing. Such a conception suits criticism very well, the latter then allotting itself the important task of discovering the Author (or its hypostases: society, history, psyche, liberty) beneath the work: when the Author has been found, the text is 'explained'-victory to the critic. (Barthes 1997a, p. 147)

Removing the Author from the work makes deciphering it futile because it simultaneously removes the idea of the Author's intention and final signified. Once the reader relinquishes this presupposition of discovering what the Author *really* meant, as if solving a complicated math problem, the reader also destroys all the limits which this assumption presupposes. The work ceases to have a single, static, "correct" interpretation, which is really not an interpretation at all, but rather a discovering of the authorial intention. Barthes' concept of the Author is closely related to its correlated concept: the work.

"The work" is the product of the Author, the space of actual words-as-signifier into which the Author places their intention, and through which the meaning-as-signified is to be grasped by the reader. Barthes' describes the work thusly:

> The work closes on a signified . . . either it is claimed to be evident and the work is then the object of a literal science, of philology, or else it is considered to be secret, ultimate, something to be sought out, and the work then falls under the scope of a hermeneutics, of an interpretation (Marxist, psychoanalytic, thematic, etc.) . . . the Author is reputed the father and owner of his work: literary science therefore teaches *respect* for the manuscript and the author's declared intentions. (Barthes 1997b, p. 158)

The work is the receptacle of the Author's meaning, and the passive reader (or consumer) is meant to search this meaning out and, such as a piece of pottery from an archeological excavation, it should be studied for clues as to how and where the author's intention

lies. Barthes depicts and problematizes the work as being either the object of scientific investigation or a sort of talisman which contains a truth for which the reader must find the correct interpretation. The search for "authorial intention" represented by such notions as "Author" and "work" posit a single, definite, and unchanging meaning within the work and treats the work as a sacred object: sanctity for the Author, deference for the reader.

Bringing attention back to the *Laozi* and the ideas surrounding its authorship, the modern debate within China concerning authorship of the *Laozi* can be traced to late 19th century and early 20th century China's National Learning (*guoxue* 國學) movement, which included trends in evidential learning (*kaoju xue* 考據學), textual exegesis (*xungu xue* 訓詁學), and "skepticism toward antiquity" (*yigu sichao* 疑古思潮). The influential thinker Zhang Dainian 張岱年 (1909–2004) summarized the positions of major thinkers active in this movement on the time during which Laozi existed or the *Laozi* was compiled as follows:

> In the 1920's skepticism towards antiquity become popular and Liang Qichao 梁啓超 put forward the idea that the *Laozi* was not a work from the Spring and Autumn period. In his History of Chinese Philosophy, Feng Youlan 馮友蘭 placed the *Laozi* after the *Mengzi*, but before the *Zhuangzi*. Qian Mu 錢穆 and Gu Jiegang 顧頡剛 went even further and placed the text after the *Zhuangzi*. Hu Shi 胡適 and Guo Moruo 郭沫若, however, persisted in the belief that Laozi the person and Confucius were of the same time period and that Laozi was older in years than Confucius . . . In the 1950's I rethought the problem of the time period in which Laozi lived and realized that The Records of the Historian could not be ignored and once again placed Confucius and Laozi in the same period, Laozi being the elder . . . (Zhang 1992, pp. 74–78)

These changes in the Chinese intellectual community, influenced by western ideas of textual exegesis, represent a move towards more scientific rigor and had a tremendous effect on how Chinese scholars conduct textual research and how they understand the intellectual history of China. Despite this increased emphasis on evidential research which led to suspicion of the *Laozi*'s authorship, the traditional belief regarding the authorship of the *Laozi* is widely maintained to this day in China, whereas in English language literature disbelief and ambivalence is more common. The reader's position on the authorship of the *Laozi*, whether that be belief in Laozi as author, disbelief, or ambivalence, often carry important implications for one's interpretation of the text.

As a result of its poetic language, a wider range of possible meanings can be read into the *Laozi* than, for example, *Common Sense* by Thomas Paine, *On the Electrodynamics of Moving Bodies* by Albert Einstein, or directions to building a cabinet bought at Ikea. Given the dearth of historical information about its authorship, this becomes more complicated by those who read it as a *work* produced by an *Author* whose original intention and meaning must be searched out. This is evidenced by the usage of "*yuanyi*" 原意 (original meaning) commonly used in Chinese literature about the *Laozi*. While belief in the existence of a wise author whose profound wisdom can be found within this work (whether that be the empirical author or Author as a web of contexts) certainly provides a sense of significance and invests it with meaning in many regards, it also leads to a number of implications which limit the reading of a text, especially when dealing with one which has been altered over many years. Placing the position of the Author atop the hierarchy of considerations forces the reader to think of their interpretation as a deciphering of the original intention of the Author. This presumption places limitations on the production of meaning and restricts the many possible interpretations to those which strictly coincide with the historical contexts of the late Spring and Autumn period (770-476 BCE). Additionally, the earliest accessible version of the *Laozi*, in this case the Guodian bamboo slips, must be valued over and above all other versions as it is the closest to the supposed urtext authored by Laozi. Such attitudes can be found in the work of two contemporary scholars of early Chinese thought, Guo Yi 郭沂 and Chen Guying 陳鼓应.

Guo Yi (b. 1962) believes that there existed an original *Laozi* text authored by Laodan which was altered over the centuries. Guo argues for the superiority of the Guodian bamboo edition over the received editions, citing 26 ways in which the Guodian edition is superior to the received edition. He pays particular attention to passages of the Mawangdui and received versions of the *Laozi* which appear to be critical of the *ru* (儒 Confucians) but which do not contain a sense of criticism in the Guodian bamboo slips, arguing that the *Laozi* should be read as compatible with the traditional Confucian ethos. The received edition and the Mawangdui silk scripts contain lines and chapters which seem to directly criticize traditional *ru* values, such as chapters 18 and 19, "when the great Dao is dispensed with, then there is humanity and righteousness" (Moeller 2007, p. 47) and "abandon humaneness and discard righteousness and the people will return to filial piety and care". (Moeller 2007, p. 49) Guo Yi argues against this reading citing the Guodian edition.

> The bamboo edition does not have pointed language against the *ru* ethical perspective, and those passages in today's edition which obviously negate the *ru* ethical perspective are either different characters from the bamboo edition or additions and subtractions . . . In today's edition chapter 19 says "Discard sageliness and get rid of wisdom" and "discard humaneness and get rid of responsibility," while in the bamboo edition it reads "discard wisdom and get rid of distinctions" and "discard artificiality and get rid of deliberation". With one character being different the thought is completely opposite. (Guo 1998, p. 51)

The existence of passages which appear to be critical of the *ru* ethos poses somewhat of a problem for those who believe that the Laozi was written in the late Spring and Autumn period by a teacher of Confucius. The older Guodian version of the *Laozi* being uncritical of the *ru* can serve as evidence for the existence of a Spring and Autumn Period author or urtext because the lack of criticism might point to the original text having been written before the *rujia* 儒家 (Confucian school) and *daojia* 道家 (Daoist school) became distinct or even oppositional schools. Reading the Guodian edition of the *Laozi* as uncritical of the *ru* is still a contentious issue, with some scholars arguing that the Mawangdui and received editions are more specifically anti-Mencian, while the Guodian edition is critical of the pre-Mencian *ru* orthodoxy. (Henricks 2000).

Such an interpretation is constructed around the belief that an author or urtext existed prior to the traditionally dated *lunyu* 論語 (Analects of Confucius), which compels the reader to disregard any interpretation which reads the *Laozi* as critical of the *ru*. The idea of accepting a single ancient author while recognizing the lack of hard evidence for this supposition (one could call it an act of rational faith) does indeed contribute to making significance of the text insofar as it imbues it with a sense of sanctity. However, the idea that the oldest edition of the text is necessarily more valuable or more "right" than later editions because it is closer in terms of language and ideas to the presumed original work dismisses later editions. This prevents the reader from accessing new meanings generated from an evolving text which has responded to new cultural stimuli and developed new language and concepts as a result. Guo Yi's belief in the Guodian edition's temporal proximity to an original urtext written by the ancient author Laozi colors his belief in the superiority of the Guodian edition. This in turn influences his interpretation such that the Guodian version must necessarily be superior to later editors which are more distant in content from the sanctity of that imbued by the original Author. It is perhaps due to the importance he places on finding the author's intention that he rejects interpretations of the *Laozi* which would place the text in some sort of hostile discourse with the *ru*.

Similar attitudes can be found in the work of another esteemed scholar, Chen Guying (b. 1935), who holds the belief that Laozi and pre-Han Daoism had many similarities with the *ru* and, such as the *ru*, wished to affirm humaneness and positively effect society. Defending this view of Laozi's original intention, he writes that chapter 19's promotion of abandoning humaneness and righteousness was an alteration by later scholars.

> The reality is, however, that the heated antagonism between competing schools of thought, particularly Daoism and Confucianism, did not appear until the

later part of the warring states period. If this chapter attacks the Confucian, and possibly Mohist, advocacy of humaneness and responsibility, as many scholars have claimed, then the original words of Laozi were altered, a theory that fits well historically. The transmitted Laozi's push against sages, humaneness, and responsibility is most likely a mid-to-late Warring States addition. The original *Laozi*, on which the Guodian editions were based, lacks such a focused statement against Kongzi (Confucius). The additions occurred in accord with popular post-Zhuangzi Daoist arguments. (Chen 2015, p. 68)

This is another example of how attachment to an Authorial intentionalist perspective of the *Laozi*, or any work that went through a historical evolution, can lead one to belief there is a more 'correct' edition amongst its various incarnations. Although both Guo Yi and Chen Guying use the same chapters of the *Laozi* for their arguments, the Chen Guying passage quoted above highlights another way in which an Authorial intentionalist perspective can limit the possible meaning found within the work.

Whereas Guo Yi's attachment to the Authorial intentionalist perspective leads to a stronger emphasis on the *elevation* of the oldest edition of the *Laozi* to which we have access (i.e., the Guodian edition), for similar reasons Chen Guying emphasizes the *depreciation* of the later editions as alterations of "the original *Laozi*, on which the Guodian editions were based". Rather than understand the Mawangdui and received editions as valuable instances of a work which emerged through the evolution of the text and from which new meaning and significance can be generated, Chen devalues them as corruptions of an ancient urtext which should be omitted because they are a few more degrees of separation away from the supposed original text written by the supremely wise Laozi. Regardless of any distinctions between Guo and Chen's approach, they lead to the same result in the prioritization of the Guodian version over others. In Chen's commentary to chapter 18 he advocates rejecting the lines found in the Mawangdui and received versions which appear to criticize the values that would become associated with traditional Confucianism on the grounds that "these phrases are not present in the Guodian Bamboo Slips version and they should be omitted. The superfluous addition of these phrases is probably the result of the influence exerted by the theories of extremist followers of Zhuangzi in the late Warring States period, who preposterously added them to the text". (Chen 2020, p. 139) Furthermore, by regarding the traditional model of Authorship as the paramount standard of interpretation, the text must necessarily be strictly constrained by a historical, cultural, and political context which renders it of little use to a contemporary reader living in a world which is historically, culturally, and politically vastly different from 6th century BCE China.

This Barthesian critique of "the Author" as applied to the *Laozi* reveals how the reader's assumption that the goal of interpreting the *Laozi* is to discover Laozi's original intent might limit the potential interpretations and possible significance generated by the reader. The elevation of the Guodian edition as the text closest to the ancient urtext forces one to interpret it in such a way that it corresponds with the cultural milieu of 6th century BCE China, the time during which the supposed author lived. It also forces one to devalue all other editions, including the Mawangdui, Heshang Gong, Wang Bi, etc. as alterations and corruptions of the original *Laozi*. Being oriented toward discovering the Author's intention might lead the reader to casually overlook or devalue the meaning inherent in these alterations, alterations which occurred because of real historical, cultural, and political events, in favor of deciphering what Laozi "really meant". Because the Author's intent is the final arbitrator of such a reader's interpretation, the depth of potential meaning found in the text as a result its evolution might be ignored by the monomaniacal seeker of the Author. While the authorial intentionalist hermeneutic position does indeed provide ways which open potential significance within the text, it is also worthwhile to recognize the ways it limits the text's plurality of meanings and reduces it to a work with a single, static interpretation placed within a specific locus of history. Given the problematic nature of searching for the Author's original intent within the work, Barthes offers a way of reading

which opens the reader up to infinite possible meanings within the text which will be referred to as the "reader-oriented Text".

### 3. The Reader-Oriented Text and Its Limits

While "the work" is related to "the Author" as that empirical space into which the Author's intention is deposited, he proposes the notion of "the Text"[3] which is related to "the reader" as "a methodological field ... *the Text is experienced only in an activity of production* ... the Text is *radically* symbolic: a work *conceived, perceived and received in its integrally symbolic nature is a* text". (Barthes 1997b, p. 159) Rather than regarding the Author as the sole supplier of meaning, Barthes shifts the focus onto the reader, who ultimately is the sole creator of meaning. Although words on the page are being read, the Text cannot be said to have started until the reader begins to *play* with relationship between signifier and signified to produce meaning. Barthes writes:

> A text is made of multiple writings, drawn from many cultures and entering into mutual relations of dialogue, parody, contestation, but there is one place where this multiplicity is focused and that place is the reader, not, as was hitherto said, the author ... the reader is without history, biography, psychology; he is simply that *someone* who holds together in a single feild all the traces by which the written text is constituted. (Barthes 1997a, p. 148)

The reader, being unrestricted by the delusion of deciphering the Author's intention, becomes open to the infinitude of meanings which can be derived from the unstructured intertextuality that occurs in the act of reading. With this newfound freedom, engagement with the text as a "work" transforms into engagement with the process of semiosis itself, or the "Text". Barthes often describes this act of reading as one which is radically detached from any attempt to read into the text an authorial and historical context, which can be described as a radical reader-oriented hermeneutic stance.[4]

In the case of the *Laozi*, such interpretations which treat it as "the Text" rather than "the work," unburdened by the exclusive interpretive standard oriented around the Author's intention, often use semantics such as grasping "the spirit of the book" (Le Guin 1998, p. 115) as opposed to finding the Author's "original meaning" and "the freest translation is often the most faithful" (Mitchell 2006, p. xii) as opposed to attempting to strictly translate the words of the Author. Despite Stephen Mitchell's belief that the ancient author Laozi did exist (Ursula Le Guin seems relatively ambivalent about this), this sort of language exposes a very different hermeneutic stance than that of the authorial intentionalist, one according to which the potential for significance of the text lies much more emphatically in the reader's contribution to it. While not quite representing a *radical* reader-oriented position, neither of these translators are able to read Chinese and thus forego any presumption of a faithful rendering of the Author's words. Furthermore, external historical, linguistic, or intertextual standards cannot be applied to the "spirit" of a text in the same way they might to the historical context of the Author, thereby allowing the reader much more leeway in depicting the "spirit of the book". Such an attitude can extend the realm of potential significance within the text far beyond the constraints of Authorial intentionalism, fostering novel connections with other non-native cultures and ideas, a fusion with modern trends and phenomena which might breathe new life into the ancient text, or the possibility of applying the text to a wider set of socio-political problems in contemporary society. This more open attitude can be found in interpretations which read the text as promoting a libertarian or anarchic approach to the political economy, (Boaz 1998; Stamatov 2014) something akin to new age spirituality, (Kohn 2019) or environmentalism (Girardot et al. 2001; Nelson 2009; Schönfeld 2014) amongst many others.[5] Some examples cited to defend such interpretations as valid include interpreting "act through non-action, then there will be nothing that is not ordered" (Moeller 2007, p. 9; translation modified) as a parallel to laissez faire ideas such as Friedrich Hayek's "spontaneous ordering," "concentrate the and attain softness" (Moeller 2007, p. 25) as teaching a form of spiritual cultivation which allows one to connect to the *dao*, and "[the sage] is able to support the nature of the ten-thousand

things" ([Moeller 2007](), p. 149; translation modified) as an exhortation for human society to be in ecological harmony with the natural world.

While these interpretations are indeed helpful in lending new significance to the text, such as the authorial intentionalist position, an excessively reader-oriented interpretation can also lead to problematic results. In diminishing, or in the case of the *radical* reader-oriented stance completely ignoring the historical, political, intellectual, and linguistic context in which the *Laozi* was compiled in favor of capturing a decontextualized "spirit of the book", the "freest translation" often becomes more of a portrait of the interpreter's own beliefs, sensibilities, and experiences than a reflection of the content of the *Laozi*. Barthes himself recognized this pitfall, emphasizing the importance of re-reading because "those who fail to reread are obliged to read the same story everywhere". ([Barthes 1974](), p. 16) Prejudiced and biased interpretations in which the interpreter *excessively* projects their own ideas and personal beliefs into the text can be seen in the use of cherry picking, highlighting chapters which validate one's reading while ignoring those parts which are unrelated or even contradict it, a common phenomenon in both academic and popular literature on the *Laozi*. The reader always carries within themself a world of experiences, biases, knowledge, culture, in short, a worldview, through which they make sense of the world and turn this 'blooming, buzzing confusion' into something able to be navigated. The unreflexive projection of these internal and prejudiced worldviews, which is to some degree unavoidable, and the corresponding diminishing of external standards of interpretation can potentially lead to the very opposite of what is intended, a reduction in the production of meaning. By creating a sort of zero-gravity interpretive environment in which no external standards tether the reader to the ground, beliefs, ideas, and modes of thinking already affirmed by the reader prior to textual engagement are merely re-established. Having external standards for what counts as a good or bad interpretation, whether it be historical, conceptual, or linguistic, helps contribute to the production of significance and meaning by challenging the reader to break out of habitual thought, a theme which ironically can be found in the *Laozi* itself (chapter 44). These standards for interpretation serve as a tool to challenge the reader, breaking them out of their own complacency and creating new realms of significance for both the text and the reader.

The disadvantages of a reader-oriented theory can further be seen from a more pragmatic perspective as, there being no standard to judge any interpretation as better or worse than another, putting it into practice would lay waste to all discourse concerning any particular text. For there to exist discourse about anything in general there must exist some agreement amongst the interlocutors on a standard of evaluation, otherwise the discourse must pivot towards what is the best standard of evaluation, which assumes yet another agreed upon standard. This is what Richard Rorty calls "normal discourse," which is the agreement on a "set of conventions about what counts as a relevant contribution, what counts as a question, what counts as having a good argument for that answer or a good criticism of it" ([Rorty 1979](), p. 320). This "normal discourse" then allows for "the sort of statement that can be agreed to be true by all participants whom the other participants count as 'rational'" ([Rorty 1979](), p. 320). Along similar lines, Stanly Fish argues that while it is through the reader's interpretation that the literary work is created, this does not invite an unchecked proliferation of interpretations because "it is interpretive communities, rather than either the text or reader, that produce meanings". ([Fish 1980](), p. 14) The idea of producing meaning outside of any discourse within a community is not only contradictory because all concepts, beliefs, and worldviews ultimately derive from discourse within a community, but it is also antithetical to all academic disciplines, especially philosophy, for which the idea of dialogue and discourse serves as a foundation. Indeed, it is even necessary for knowledge of the physical sciences which, as Kuhn points out, exists within the matrix of scientific communities ([Kuhn 2012]()).

### 4. Intention of the Text and the Production of Meaning

We are faced with two extremes: the dogmatic authorial intentionalist perspective and the radical reader-oriented perspective. The strengths each of these particular perspectives cannot be denied, but they carry with them a number of limiting factors and disadvantages which makes an uncritical attachment to them untenable. We are thus led to take a stance which avoids adherence to the specter of the Author while having some sort of grounding on which discourse and interpretations of the text can exist. Umberto Eco attempts to create a theory of semiosis which does just this by allowing for numerous interpretations of any given text while also providing external standards to which any given interpretation might be subjected. While influenced by thinkers who extended conceptions of the field of semiosis and particularly by Barthes himself, he argued that there must exist some standards for interpretation on the grounds that "the notion of unlimited semiosis does not lead to the conclusion that interpretation has no criteria. To say that interpretation (as the basic feature of semiosis) is potentially unlimited does not mean that interpretation has no object and that it 'riverruns' merely for its own sake" (Eco 1994, p. 6). Regarding the opposition between the authorial intentionalist theory and the radical reader-oriented theory, Eco points out "one can object that the only alternative to a radical reader-oriented theory of interpretation is the one extolled by those who say that the only valid interpretation aims at finding the original intention of the author . . . there is a third possibility. There is an *intention of the text*" (Eco 2004, p. 25).

The "intention of the text" is the interplay between the model reader and the model author. The model reader can be thought of as the act of reading in which conjectures are made about the intention of the text, a sort of reading in which infinite conjectures are allowed. The model reader's activity and productivity lie in this act of making conjectures. These conjectures, however, are limited by the model author (as opposed to the empirical one, which Barthes refers to as the Author). The model author is an interpretation (one of many possible interpretations) that coincides with the intention of the text. Regarding the relationship between model reader, model author, and intention of the text, Eco says "more than a parameter to use in order to validate the interpretation, the text is an object that the interpretation builds up in the course of the circular effort of validating itself on the basis of what it makes up as its result" (Eco 1994, p. 59).

But how does one know which model author, which is to say, which interpretations, coincide with the intention of the text? Eco provides the standard for this judgement as "any interpretation given of a certain portion of a text can be accepted if it is confirmed and must be rejected if it is challenged by another portion of the same text. In this sense, the internal textual coherence controls the otherwise uncontrollable drives of the reader" (Eco 1994, p. 59). Here we see a simple, yet powerful concept in textual theory which reigns in the excessive freedom of the Barthesian reader: coherence. He further develops the concept of textual coherence with a more concrete standard for what is considered textually coherent, that of isotopy, an umbrella term which includes many different types of isotopies, but all exhibiting a common trait which is defined as "constancy in going in a direction that a text exhibits when submitted to rules of interpretative coherence" (Eco 1980, p. 153).

The many different types of isotopies widen the limits of possible acceptable interpretations, i.e., interpretations with high levels of interpretive coherence. These limits are simultaneously restricted by the language of text, to which these isotopes must refer. Here Eco's standards for textual coherence can once more be found in the interplay between author and reader, or what he refers to as "interpretive cooperation", which he defines as:

> an act in the course of which the reader of a text, through successive abductive inferences, proposed topics, ways of reading, and hypotheses of coherence, on the basis of suitable encyclopedic competence . . . determined by the nature of the text. By the "nature" of the text I mean what an interpreter can actualize on the basis of a given Linear Manifestation, having recourse to the encyclopedic competence toward which the text itself orients it Model Reader. (Eco 1980, p. 154)

The various ways in which different isotopies can be read into a text, creating a plurality of interpretations, provides semiotic freedom, while the "nature of the text" and a general standard of coherence tethers the interpretation to the text and provides a sort of semiotic foundation.

Freeing one's interpretation from the search for the intention of the empirical author as the paramount goal of reading, yet still holding one's interpretation to the standard of textual coherence (which may include philological, historical, or conceptual coherence) may open the potential meaning produced when engaging with each particular edition of the *Laozi* while still being grounded in possible discourse. If this model is extended further to intertextual coherence with, let us say a tradition of texts, the evolution of the *Laozi* may likewise be opened up to greater potential meaning. Similar hermeneutic positions can also be found within Chinese intellectual history and the commentarial tradition of the *Laozi* in particular. The legitimacy of interpretations within Chinese commentarial traditions are established according to whether or not the ideas therein are *tong* 通 (continuity, connection), which implies internal coherence or continuity as well as continuity within the larger tradition. Within the Chinese commentarial tradition "the goal of 'explaining the classics' is achieving *tong*, going from character, to word, to sentence, to paragraph, to chapter, and extending to the entire internal text and its intertextual relationships, gradually generating a network of connections and organic system" (Bao 2015, p. 3). Likewise, Rudolph G. Wagner describes Wang Bi's hermeneutic position in regard to the existence of Laozi in similar terms to those of Eco's, writing "the assumption of the *Laozi's* being written by an author called Laozi is an assumption about the philosophic homogeneity of this text" (Wagner 2000, p. 120). It should thus not be surprising that the notion of hermeneutic coherence, whether it be linguistic, conceptual, or historic, can also be found in contemporary Chinese scholarship on the *Laozi* in such a way that it allows for equal estimation for the various editions of the *Laozi* as opposed to the authorial intentionalist, while still holding those interpretations to external standards.

An example of such a reading of the *Laozi* can be found in that of another esteemed scholar, Liu Xiaogan 刘笑敢. Liu believes that there did exist an urtext from which the other editions eventually developed and also agrees with Chen Guying and Guo Yi about the Guodian edition of the *Laozi* being at least much less critical of the *ru*, perhaps even compatible. As with Chen and Guo, he regards the attitude towards *ruist* values to be one of the major differences between the Guodian edition and later versions.

> As for the relationship between Daoists and Ru, a popular perspective is that the two are at battle . . . In fact, before the Song Dynasty there was not any textual foundation for mutual antagonism or criticism between the Ru and Daoists . . . the sentences "eliminate sageliness, get rid of wisdom" and "eliminate benevolence, get rid of righteousness" found within the received edition of the Laozi is the result of language being inserted into the text by later editors. This sort of insertion is the most serious kind of tampering done within the evolution of the various editions of the Laozi, but it is not completely without basis. In any case, the received edition clearly projects and intensifies a critical attitude towards Confucianism . . . the hierarchy and position of Laozi's *dao* and *de* is higher than Confucian morality, but does not simply deny or replace Confucian morality. (Liu 2016)

An important distinction between Liu and Chen/Guo's positions, however, is that when Liu discusses the differences between the Guodian edition and received edition, he does not elevate the status of the Guodian edition as being "more correct" or "more authentic," nor does he denounce the later received editions as corruptions and therefore less meaningful than the earlier editions. He seems to recognize the alterations made by editors and compilers of the later editions without making any value judgement.

Rather than dismissing the alterations as deviations from the *original* manuscript, he uses the evolution of the various manuscripts as something which can potentially provide meaning. Just as Eco provided ways of testing coherence in individual texts in the form of isotopes, Liu provides ways of testing coherence in the evolution of a text in the form of

linguistic assimilation and conceptual focusing, which he defines as "the general tendency of editors of the Laozi to replace some words, phrases, or passages with common terms or patterns according to their understanding of the message and style of the text" and "processes designed to bring out the intellectual insights and key concepts of the *Laozi*" (Liu 2003, p. 338). Conceptual focusing in particular indicates that Liu sees these alterations to the text as contributing to valid and valuable concepts.

Making a much more explicit statement on his stance regarding the search for Authorial intention, he writes:

> One may think that the editors and collators believed that their alterations were restoring the original version and improving upon the extant versions. However, the bamboo and silk versions give evidence that the earliest versions were not as logical and coherent as later scholars thought and wished. Approximating the original or earliest texts did not necessarily accord with the goal of improving the extant editions or creating the best version. The received versions show improvement in that they have a more regulated pattern and style than the antique versions, but they are not closer to the antique versions. Likewise, modern textual studies may improve ancient texts according to modern logic and assumptions, but they do not necessarily preserve the intent and style of the original or earliest versions. (Liu 2003, p. 381)

He regards textual coherence according to his standards of linguistic assimilation and conceptual focusing as a more valuable contribution to the received versions of the text than would have been attempting to simply make it as similar to the original or earliest versions as possible. It seems that in his interpretation of the evolution of the *Laozi*, Liu, much in the same way as Eco, navigates between the Scylla of searching for authorial intent and the Charybdis of an ungrounded reading of the text. Such an interpretation opens the field of possible meaning to include the evolution of the texts, which in turn reveals the possible, and now legitimate, significance that can be found in each individual edition in their own particular way. This interpretation is quite distinct from those of Chen Guying and Guo Yi, but this distinction all stems from a very subtle difference in hermeneutic stances regarding authorial intention.

## 5. Conclusions

The debate regarding the existence of Laozi the author is perhaps an example of misplaced energy for those on both sides of the debate, particularly those working on reading the *Laozi* from a philosophic perspective. There is no doubt that any advancements or findings on the historicity of Laozi-the-person would be exciting, but those reading the *Laozi* from a philosophic or literary perspective would be better suited considering how the perceived authorship of the *Laozi* influences their *interpretation* of the text. The intention of this paper has not been to prove any one interpretation over and above others, but is meant to bring attention to how any interpretation which lacks a critical, and thus reflexive attitude toward one's own hermeneutic position might easily succumb to problematic elements which exist for any position whatsoever. Chen Guying and Guo Yi's positions are certainly not *dogmatically* authorial intentionalist, nor are Mitchell and Le Guin's translations mere flights of fancy, but a lack of critical awareness of one's own hermeneutic position can lead to problematic results, such as an adherence to the Author's "original meaning" for the former or reading into the text a decontextualized, borderline mystical "spirit of the book" for the latter. Without a critical awareness of the hermeneutic principle according to which one interprets a text, even the principle of coherence could lead to problematic results. Whereas Liu Xiaogan's reflexive awareness of his use of the principles of linguistic assimilation and conceptual focusing allows him to point out in what ways the later editions were a continuation of aspects of the Guodian edition, yet distinct in their own right, one can easily imagine an uncritical interpreter blindly and dogmatically affirming the coherence of all editions expressing the same "spirit" of Laozi, thereby overlooking each version's unique contributions.



Conscious recognition and critical awareness of the utilization in one's interpretation of hermeneutic concepts such as authorial intention, textual coherence, "*tong*" 通 (continuity), or many possible others not only help avoid potentially problematic pitfalls, but also open new paths of interpretation, new ways of meaning, and new roles for the reader to take on. At the expense of withdrawing into naval gazing, discourse on hermeneutic principles which guide our interpretations and translations of the *Laozi* might prove to be a great source of insight, scholarship, and inspiration. For example, to further use the notion of coherence as a hermeneutic standard, a concept such as "coherence validity," similar to Lee J. Cronbach and Paul E. Meehl's explication of construct validity, (Cronbach and Meehl 1955) could be used as a general principle for the interpretation of the *Laozi*, amongst other texts. While the act of interpretation is not the sort of thing to which a hard and fast scientific method can be ascribed, discourse surrounding a notion such as coherence validity might produce new perspectives on how the already existing wealth of scholarship on the historical, linguistic, and conceptual context of the *Laozi* are related to one another in novel and creative ways, revealing new networks of significance within the text(s). This would perhaps represent a shift in focus from *what* the Author meant to *how* the text means.

**Funding:** East China Normal University; East China Normal University Institute for Wisdom in China and Department of Philosophy; 華東師範大學中國智慧研究院暨哲学系; Project Number: QN2021137001L; Shanghai, China.

**Institutional Review Board Statement:** Not Applicable.

**Informed Consent Statement:** Not Applicable.

**Data Availability Statement:** Not Applicable.

**Conflicts of Interest:** The author declares no conflict of interest.

## Notes

[1] Barthes is used to represent a trend of 20th century thinkers and schools of thought which resisted the "intentional fallacy," including T.S. Eliot, Stanly Fish, and, as will be discussed later in the paper, Umberto Eco. While many of the thinkers who represent the resistance to Authorial intent have literary works like poetry, novels, or plays in mind, similar arguments can be and have been extended to the authorship of works of philosophy (Deleuze 1985) and even to history itself. (White 1966) Problematizing Authorial intent in the *Laozi* is also defendable considering the text's literary and poetic nature.

[2] The Author is capitalized to distinguish it from the "empirical author" as the constellation of contexts which orient the work's supposed intention.

[3] Like "Author", "Text" is capitalized to distinguish it as a "methodological field" and "activity of production" rather than "text" as a work containing written text.

[4] It should be noted that this more "radical" position might be read within Barthes' work as a gesture to merely problematize the notion of Authorial intention and reveal the significance of intertextuality in the act of semiosis rather than an outright rejection of historical contextualization. This more radical position is likewise being use in this paper for the same purpose while also problematizing the position itself.

[5] The examples cited here are not necessarily examples of the *radical* reader-oriented hermeneutic as they are certainly more thorough in terms of historical contextualization. The tomb in which the Mawangdui version, for example, was found also included the *Daoyin tu* 導引圖 manuscript containing illustrations of a form of Qigong spiritual cultivation practice, providing some historical evidence for Kohn's spiritualist interpretation. Such interpretations might be considered merely reader-oriented as compared to that of the strictly Authorial intentionalist.

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
