# Peer review of "The Hierarchy of Authorship in the Hermeneutics of the Daodejing"

_religions, doi:10.3390/rel13050433_

Round 1
Reviewer 1 Report
The author could reference Paul J. D'Ambrosio's article "Approaches to ethics in the Laozi" (Philosophy Compass 17(2), February 2022). The article elaborates on the controversy around what type of ethics the Laozi could be promoting. It could thus further enrich the author's discussion concerning diverse interpretations of the Laozi and their possible underpinnings.
Author Response
Reviewer,
I very much appreciate your comments. I will take them into consideration in revising the draft.
Best regards
Reviewer 2 Report
This is an excellent paper, which shows how fruitfully modern Western hermeneutic and semiotic theories can be applied to the study of ancient Chinese texts. The discussion is argued very clearly and convincingly, and does not only illuminate different approaches to the reading of the Daode jing, but also proposes new ways of making meaning in the Daode jing.
I really enjoyed reading this paper and it definitely deserves to be published.
There are a few typos, which should be eliminated before print:
lines 22-23: authorship of the Laozi's (eliminate 's)
line 30: pg.35 space missing
line 34-35: Laozi was supposedly existed should be Laozi supposedly existed (or Laozi was supposed to exist).
line 67, not a typo but a suggestion: "because of our inaccessibility to the empirical author's inner thought" might be better as " because of our lack of access to the empirical author's inner thought" (or else " because of the empirical author's inner thought's inaccessibility to us." )
line 101 "treats the work as if a sacred object" might be better as " treats the work as a sacred object" (or as if it was a sacred object)
line 184: "as critical the ru" should be "as critical of the ru".
line 222-223: "depreceiation" should be depreciation
Author Response
Reviewer,
Thank you very much for your comments and the examples/line numbers for the typos found in the paper. Your recommendations for alterations were extremely helpful as well.
Best regards,